# What do part-time employees in Japanese chain restaurants talk about when dissatisfied? Applying Structural Topic Modeling to employee reviews

**Hiroki Takahashi** *

Institute of Systems and Information Engineering, University of Tsukuba, Tsukuba, Ibaraki, Japan

* takahashi.hiroki.gn@u.tsukuba.ac.jp

## Abstract

While part-time employees constitute the primary workforce in the chain restaurant industry, their retention has become crucial in developed countries, especially Japan, due to labor shortages resulting from the declining birthrate and aging population. Analyzing employee reviews is an effective method for understanding factors that decrease employee satisfaction. However, while many analyses are focusing on full-time employees, there is insufficient analysis focusing on part-time employees, whose employment status and motivations differ from those of full-time employees. This study employs a Structural Topic Model to correlate latent topics from 4511 online text reviews with a 5-point scale of part-time employee satisfaction scores in Japanese chain restaurants. The study identifies 20 topics, including management systems and key employee interests. Especially digital communication and interview processes frequently appeared when satisfaction was low, which are unique to part-time employees in chain restaurants and had been overlooked in previous analyses. Further analysis links 20 topics to four 5-point scale HRM metrics (compensation satisfaction, workplace environment, motivation, and interpersonal relationships), enabling deeper analysis of the relationships between topics and HRM metrics. These insights contribute to the development of strategies to enhance part-time employee satisfaction in chain restaurants.

## Introduction

Chain restaurants, such as McDonald's and Domino's Pizza, follow operational standards set by their corporate headquarters to ensure consistent service delivery to customers. In Japan, chain restaurants like Yoshinoya and Ippudo comprise over 30% of the industry [1]. These establishments maintain uniform service quality across their locations, heavily relying on frontline employees, primarily part-time workers, for direct customer interaction and brand representation. Part-time workers form a vital segment in the restaurant workforce; for instance, in the United Kingdom, they represent 54.7% of restaurant employees [2], and in

**Data Availability Statement:** The employee review data necessary for reproducing this study was obtained from the "Minhyo Review Dataset"

(https://doi.org/10.32130/idr.16.1), provided by meisterstudio, Inc. via the IDR Dataset Service of the National Institute of Informatics. To obtain the review data for reproducing this study, one first applies for and receives permission from the Informatics Research Data Repository of the National Institute of Informatics (https://www.nii.ac.jp/dsc/idr/minhyo/). Then, anyone can access the data by processing the downloaded dataset according to the method indicated in the "Data collection" in the Methodology section of this paper.

**Funding:** This study was supported by grants from the Japan Society for the Promotion of Science (KAKENHI 23K17003). The funders had no role in study design, data collection and analysis, decision to publish, or preparation of the manuscript.

**Competing interests:** The author has declared that no competing interests exist.

Japan's hospitality and food service sectors, they constitute 63.1% of the total workforce [3]. In this paper, 'part-time workers' refers to 'non-regular employees (非正規職員)'[4] as classified by Japan's Statistics Bureau [3].

However, advanced economies, including Japan, face critical labor shortages in the service industry due to a dwindling workforce [5, 6]. This shortage poses a serious threat to service quality. Declining employee satisfaction not only impacts their turnover intentions [7] but also affects customer satisfaction, as indicated by the service profit chain theory [8].

This study examines online review data from part-time employees in Japanese chain restaurants to gain insights into employee satisfaction. Traditional surveys often fail to capture the complexity and nuances of employee satisfaction, leading to potential biases and limited scope [9]. In contrast, qualitative data from reviews can reveal aspects of satisfaction or dissatisfaction that structured surveys might miss [10]. Such reviews provide unfiltered insights into employees' thoughts and feelings, including reasons for leaving their jobs, which are challenging to obtain through internal company surveys post-resignation.

Using a Structural Topic Model (STM) [11], this study analyzes text-based reviews with 1-to-5 scale human resource management (HRM) related ratings (e.g., employee satisfaction, compensation, workplace environment, motivation, and interpersonal relations) by part-time employees in Japanese chain restaurants. The aim is to explore the relationship between topics in part-time job reviews and low satisfaction, leveraging STM's ability to integrate and analyze these elements.

The study has two objectives: first, to identify key topics emphasized by part-time employees in the chain restaurant industry regarding their job satisfaction; and second, to determine which topics become more prominent during periods of low HRM scores. This will help understand the issues that need prioritizing in response strategies by the corporate headquarters. This research enhances our understanding of employee satisfaction in the service sector through the analysis of unstructured data from part-time employee reviews. It contributes significantly to hospitality, service, and management research fields. Additionally, applying STM to real-world Japanese employee review data is a noteworthy advancement, demonstrating the utility of topic analysis in understanding employee perspectives within the chain restaurant industry. The findings offer not only academic significance but also practical implications for service industry managers aiming to improve part-time employee satisfaction and retention.

## Literature review

### Restaurant chain employees

Employee satisfaction in chain restaurants is significantly shaped by their organizational structures. Chain restaurants, characterized by common ownership and uniformity in quality and services across multiple locations [12], centralize management at the headquarters. This centralization allows stores to focus on service provision, distinguishing them from independent restaurants which lack such supervisory presence. Typically, a store manager, responsible for overseeing store employees, manages each location. The headquarters staff, although overseeing these managers and handling related tasks, usually have limited direct interaction with part-time employees. Consequently, issues related to supervision and interpersonal relationships tend to arise more frequently among store managers and employees, rather than with headquarters staff.

Employee job satisfaction in chain restaurants also hinges on the experiences of part-time staff. These roles often contrasted with full-time positions, show significant differences in aspects such as job satisfaction, organizational commitment, turnover intentions, and morale [13, 14]. Furthermore, perceptions about part-time work [15], as well as challenges in career

progression, including limited promotion opportunities and wage disparities, are noteworthy concerns [16].

Frontline employees represent a pivotal aspect in the dynamics of service industry chain stores. Their direct interaction with customers places them at the heart of the service profit chain concept, as outlined by Heskett et al. (1994): [8]: employee satisfaction ultimately influences customer satisfaction. Additionally, the reverse influence of customers on employees is significant. Frey et al. (2013) [17] demonstrate that positive customer interactions can significantly enhance employee satisfaction. On the flip side, encounters with uncivil customers have been shown to adversely affect employee morale [18]. Understanding this bidirectional relationship is crucial for improving both customer service and employee job satisfaction in these settings.

Specifically focusing on part-time restaurant employees, Sobaih et al. (2011) [2] used interviews and surveys to identify areas where part-time employees are less satisfied compared to full-time staff: (1) lack of pre-job briefing, (2) number of working hours and schedule, (3) training, (4) pay and benefits, (5) promotion opportunities, (6) operational involvement, and (7) performance evaluation procedures. However, prior studies have not conclusively determined which of these aspects part-time employees in restaurants prioritize most, are most likely to vocalize, and managers should therefore prioritize for improvement. Therefore, this study attempts to gain a comprehensive understanding of the topics most frequently mentioned by part-time employees when expressing dissatisfaction, using reviews and HRM scores.

## Utilization of online employee reviews

While methods like questionnaires, interviews, and tests have been used historically to gauge job satisfaction [19], employee surveys have consistently been the primary tool for evaluation in most studies [20]. However, these surveys often face limitations such as infrequency and small sample sizes [21]. Furthermore, surveys, conducted through interviews or questionnaires by researchers or HR teams, may not always yield honest responses due to employees feeling pressured by senior management [22].

Analyzing online employee reviews offers a uniquely beneficial approach compared to traditional methods. These unstructured reviews provide insights into areas often missed by surveys [7]. Their immediacy and broader scope are essential for capturing real-time market responses and generalizing findings [7]. Additionally, reviews are typically posted voluntarily and anonymously on platforms, likely reducing any pressure employees might feel from superiors or researchers [22].

The effectiveness of analyzing online employee reviews makes it an ideal method for assessing part-time employee satisfaction in service industries. Recent research trends reflect a growing reliance on this approach. Studies are increasingly using data from employee-written reviews on platforms to evaluate organizational satisfaction levels [9, 22, 23]. For instance, Sainju et al. (2021) [9] analyzed over 682,176 reviews from Fortune 50 companies on Indeed. com, uncovering key insights into management and monetary benefits' impact on employee satisfaction and turnover, with notable industry differences. Similarly, Jung and Suh (2019) [23] examined over 35,063 reviews from jobplanet.co.kr, identifying essential job satisfaction factors beyond traditional HRM surveys, including Project, Software Development, and Marketing. In hospitality fields, Stamolampros et al. (2019) [7] conducted an analysis of employee reviews on Glassdoor for entities including restaurants, hotels, and tourism. Their findings indicate a correlation between employee satisfaction levels and prevalent topics: higher

satisfaction aligns with frequent mentions of Career Opportunities and Work Environment, while lower satisfaction correlates with discussions centered on Employee Perks and Compensation.

Despite its potential, this approach's application, especially in chain restaurants' part-time employee satisfaction, remains underexplored.

## Structural Topic Model

The proliferation of data from digital platforms necessitates advanced text analysis methods like topic modeling. STM offers significant advantages over traditional topic models as it allows researchers to integrate arbitrary document information in the form of covariates for estimating per-document topic distributions (topic prevalence) and per-topic word distributions (topic content) [10, 24]. It uniquely estimates the impact of document metadata on latent topics during modeling. In hospitality research, STM has predominantly been applied to customer reviews, providing intricate insights into customer satisfaction [10, 25, 26]. For example, Hu et al. (2019) [25] analyzed 27,864 New York City hotel reviews using STM, identifying 10 dominant topics in negative reviews. Similarly, Ding et al. (2020) [10] focused on Malaysian Airbnb reviews, revealing a distinct preference for appearance and location among Malaysian users. In contrast, the application of STM to employee reviews remains relatively rare. Notable exceptions are the work of Sainju et al. (2021) [9] and Stamolampros et al. [7] which are mentioned in Section. These studies employed STM to analyze employee reviews, effectively differentiating topics between current and former employees based on job status metadata.

This study expects that employing STM will facilitate the categorization of topics, taking into account the influence of HRM factor ratings, time, and service industry categories. Furthermore, it aims to uncover associations between these topics and HRM factors, thereby offering a deeper understanding of the thematic underpinnings of part-time job discussions.

## Methodology

### Data collection

This study used data from 'Minhyo', a prominent Japanese review platform, provided by meisterstudio, Inc. via IDR Dataset Service of National Institute of Informatics, comprising approximately 160,000 reviews across 4,700 products and services from January 16, 2017, to July 12, 2022 [27]. For this research, the dataset of reviews by part-time employees in chain restaurants (e.g., cafes, family restaurants, fast food, izakaya, and other establishments) is used. The dataset includes textual reviews and five HRM ratings (総合満足度(*Employee satisfaction*), 給料 (*Compensation satisfaction*), 働きやすさ (*Workplace environment*), やりがい (*Motivation*), and 人間関係(*Interpersonal relationships*)) on a 1 to 5 scale. For our analysis, we selected reviews with numerical ratings, excluding those marked 'NA' since 2019 due to introducing "NA" as a rating option in 2019.

The dataset contains 4,511 reviews with ratings, as summarized in Table 1. A significant portion (50.4%) relates to family restaurants. Average ratings are generally low, for instance, *Employee satisfaction* at 1.790 and *Workplace environment* at 1.897, suggesting a prevalence of negative feedback. This aligns with the findings of previous research that suggest a tendency for extreme opinions to be more frequently posted [28]. It seems to indicate the presence of a self-selection bias, where moderate opinions are less likely to be posted [29]. Such a pattern suggests that the most frequently mentioned topics in lower-scoring reviews likely reflect the most pressing issues for employees, aligning well with the study's objectives. The correlation

**Table 1. Descriptive statistics.**

|  | M(SD) / n(%) |
|---|---|
|  | N = 4,511 |
| Employee satisfaction | 1.790 (1.155) |
| Compensation satisfaction | 2.319 (1.202) |
| Workplace environment | 1.897 (1.206) |
| Motivation | 2.265 (1.349) |
| Interpersonal relationships | 2.229 (1.366) |
| Category |  |
| Cafe | 589 (13.1%) |
| Family restaurant | 2,275 (50.4%) |
| Fast food restaurant | 488 (10.8%) |
| Izakaya | 98 (2.2%) |
| Other restaurant | 1,061 (23.5%) |

**Table 2. Correlation matrix.**

| Variables | (1) | (2) | (3) | (4) | (5) |
|---|---|---|---|---|---|
| (1) Employee satisfaction | 1 |  |  |  |  |
| (2) Compensation satisfaction | 0.43 | 1 |  |  |  |
| (3) Workplace environment | 0.78 | 0.39 | 1 |  |  |
| (4) Motivation | 0.66 | 0.39 | 0.66 | 1 |  |
| (5) Interpersonal relationships | 0.67 | 0.31 | 0.7 | 0.58 | 1 |

matrix in Table 2 revealed the highest correlation between *Employee satisfaction* and *Workplace environment*. Particularly, *Employee Satisfaction* correlates with the other four HRM ratings, necessitating separate consideration, as mentioned in Section.

## Text preparation

To conduct topic analysis on the review texts, it is essential first to create a corpus from the words that make up the reviews and then vectorize the text to establish topic models. Building on existing protocols in Japanese text analysis, this study undertook a specific preprocessing routine for these texts. Since Japanese sentences do not separate words with spaces, segmentation, and morphological analysis became crucial steps. For segmentation, MeCab [30] was employed through RMeCab [31], while mecab-ipadic-neologd [32] was used for morphological analysis. This study focused on terms categorized as either "形容詞" ("adjective"), "名詞" ("noun"), "動詞" ("verb"), or "形容動詞" ("adjectival noun"), and was further refined to subcategories like "一般" ("general"), "自立" ("independent"), "サ変接続" ("irregular verbs"), "形容動詞語幹" ("adjectival stems"), "ナイ形容詞語幹" ("negative adjectival stems"), and "固有名詞" ("proper nouns"), based on previous research on Japanese text analysis [33]. Additionally, the author removed Japanese stop words [34] based on recommendations from prior research [35] and also excluded additional meaningless words, for example, "人" ("person"), "思う" ("think"), "言う" ("say"). Furthermore, the author excludes proper nouns that could directly indicate specific companies. Finally, terms that appeared less than five times in the dataset were excluded.

## Covariates setup

This study analyzes the relationship between part-time job ratings and the prevalence of topics that emerge in reviews. Utilizing STM allows considering document-level (in this context, review-level) metadata in a regression formula for estimating topic prevalence. This same approach facilitates covariate analysis, enabling the testing of coefficients for each variable. In this study, we used two models of prevalence regression:

$$
\begin{aligned}
prevalence_{d,k} = & \; \beta_0 + \beta_{ES} \times ES_d + \beta_{days} \times s(days_d) \\
& + \beta_{category} \times category_d + \epsilon_d
\end{aligned}
\qquad \text{(Model 1)}
$$

$$
\begin{aligned}
prevalence_{d,k} = & \; \beta_0 + \beta_{CS} \times CS_d + \beta_{WE} \times WE_d + \beta_M \times M_d + \beta_{IR} \times IR_d \\
& + \beta_{days} \times s(days_d) + \beta_{category} \times category_d + \epsilon_d
\end{aligned}
\qquad \text{(Model 2)}
$$

Here, $d$ is the index for the review, and $k$ is the index for the topic. $ES_d$, $CS_d$, $WE_d$, $M_d$, and $IR_d$ are the rating scores of *Employee satisfaction*, *Compensation satisfaction*, *Workplace environment*, *Motivation*, and *Interpersonal relationships* for review $d$, and $\beta_{ES}$, $\beta_{CS}$, $\beta_{WE}$, $\beta_M$, and $\beta_{IR}$ are the coefficients for the ratings. $days_d$ refers to the numerical date value for review $d$, which was converted to a numerical variable (e.g., "2017-07-16" to "1") to examine the time effect on the change in topic prevalence. $s(days_d)$ represents a spline transformation of $days_d$ in 10 dimensions, and $\beta_{days}$ is the corresponding coefficient. $category_d$ is a $1 \times 12$ dummy variable vector for the categories of review $d$, and $\beta_{category}$ is its coefficient vector.

Model 1 examines the correlation between topic frequency and *Employee satisfaction*. In contrast, Model 2 explores the relationship between topics and a linear combination of four HRM ratings (e.g., *Compensation satisfaction*, *Workplace environment*, *Motivation*, and *Interpersonal relationships*) for a more comprehensive analysis of employee satisfaction. Despite separate topic estimations for both models, no significant differences were found in the estimated topics and their associated words. Consequently, the subsequent analysis of words constituting the topics in Sections 0 and 0 will focus on the results from Model 1.

## Topic number estimation

This study utilized the approach involved in scrutinizing metrics for Semantic Coherence and Exclusivity to select the adequate number of topics, as suggested by prior research [9–11]. Semantic Coherence measures the internal cohesion of a topic based on the frequency of co-occurrence of its most probable words. Exclusivity assesses a topic's distinctiveness by comparing the distribution of its high-probability words to those in other topics. Roberts et al. (2019) [11] simultaneously advocate evaluating topic quality using both metrics. Fig 1 shows that no single-topic model holds a dominant position up to 20 topics; as the number of topics increases, Semantic Coherence generally decreases while Exclusivity rises. However, when the number of topics exceeds 20, Exclusivity drops. Consequently, around 19 topics were considered optimal. Next, topic estimation was conducted for each set from 17 to 22 topics, and the estimated topics were examined in detail. Considering the results, the interpretability of the topics, and prior research using STM [9, 10], the author chose 20 as the optimal number of topics.

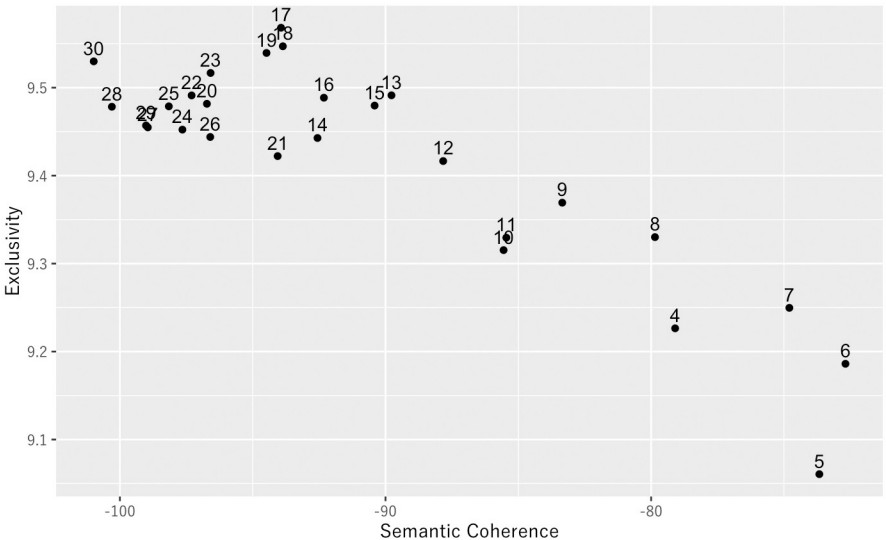

**Fig 1. Relationship between estimated Semantic Coherence and Exclusivity values for each topic count in the topic model.** Higher values towards the top right of the graph indicate a better model.

# Findings

## Topic summary and labeling

Table 3 presents 20 topics identified through topic modeling, along with their corresponding labels in Model 1. The result of Model 2 is in S1 Table. The table includes the list of Top words and FREX words that are listed in order of their highest probabilities in each topic. Top words represent words with a high probability of occurrence in each topic, while FREX words indicate words with high scores when weighted considering how exclusive they are to the topics [11].

These high-probability words were initially translated into English from Japanese by ChatGPT (gpt-4) [36] and subsequently refined by the author for enhanced interpretability. For the labeling process, the author employed ChatGPT (gpt-4) to generate preliminary labels, taking into account these high-probability words and aligning the labels with human resource management contexts. The author then reviewed and revised these labels for further clarity and relevance.

Common topics identified in general employee satisfaction reviews [7, 9] include wages (Labor Cost Reduction (Topic #8), Payroll and Interview (Topic #16), Low Wage (Topic #18)), workplace environment (Workplace Relations (Topic #2)), sense of achievement (Job Fulfillment (Topic #3)), shifts (Shift Flexibility (Topic #4)), job stress (Work Stress (Topic #11)), customer interaction (Customer Interaction (Topic #13)), and colleagues (Workplace Relations (Topic #2)). These topics are not limited to part-time workers in chain restaurants but are generally discussed by employees in prior research. However, topics like employee benefits, work-life balance, and career advancement, which appeared in other studies of employee reviews [7, 9], did not emerge here, likely because they are less relevant to part-time workers. Additionally, topics such as promotion opportunities, which were mentioned in previous studies investigating part-time employees in restaurants [2], were not observed in this research. It is possible that many part-time employees in restaurants do not expect promotion, considering their jobs as temporary.

**Table 3. Topics and labels.**

| Topic # | Topic Label | Top Words | FREX Words | Topic Proportions |
|---|---|---|---|---|
| 1 | Store Manager | 店長(store manager), パート(part-time), 社員(employee), 辞める(quit), 店(store) | 店長(store manager), 機嫌(mood), 気に入る(like), パワハラ(power harassment), 悪口(insult) | 8.92% |
| 2 | Workplace Relations | 店舗(store), 多い(many), 良い(good), 人間関係(human relations), 悪い(bad) | 仲(relationship), よる(depend), 客層(clientele), 同士(colleagues), 雰囲気(atmosphere) | 8.20% |
| 3 | Job Fulfillment | 仕事(job), 楽しい(fun), 出来る(can do), やりがい(challenge), 感じる(feel) | やりがい(challenge), 楽しい(fun), 向く(suitable), 感じる(feel), 成長(growth) | 7.63% |
| 4 | Shift Flexibility | シフト(shift), 入れる(insert), 休み(holiday), 希望(wish), 週(week) | シフト(shift), 予定(schedule), 休み(holiday), 希望(wish), 週(week) | 7.31% |
| 5 | Training Process | 教える(teach), 研修(training), 分かる(understand), わかる(understand), 先輩(senior) | 丁寧(polite), 教える(teach), 教わる(learn), 研修(training), 初日(first day) | 7.11% |
| 6 | Student Part-Time Jobs | バイト(part-time job), 高校生(high school student), 優しい(kind), 始める(start), 辞める(quit) | バイト(part-time job), 大学生(college student), 高校生(high school student), 始める(start), 友達(friend) | 7.03% |
| 7 | Newcomer Integration | 辞める(quit), 新人(newcomer), スタッフ(staff), 仕事(job), ベテラン(veteran) | スタッフ(staff), ベテラン(veteran), オープニング(opening), 新人(newcomer), 言い方(way of speaking) | 5.93% |
| 8 | Labor Cost Reduction | 時給(hourly wage), 休憩(break), 店(store), 上がる(increase), ワンオペ(single operation) | 人件費(labor cost), ワンオペ(single operation), 削減(cut), 削る(trim), 営業(operation) | 5.91% |
| 9 | Learning | 覚える(remember), メニュー(menu), 多い(many), レジ(register), 大変(difficult) | ドリンク(drink), メニュー(menu), 覚える(remember), カフェ(cafe), フード(food) | 5.02% |
| 10 | Work Area Dynamics | キッチン(kitchen), ホール(hall), 忙しい(busy), 料理(cooking), フロア(floor) | キッチン(kitchen), ホール(hall), 料理(cooking), 案内(guide), フロア(floor) | 4.81% |
| 11 | Work Stress | やめる(quit), きつい(hard), メンタル(mental), 精神的(mental), 怒る(angry) | やめる(quit), メンタル(mental), 精神的(mental), きつい(hard), 後悔(regret) | 4.79% |
| 12 | Company | 会社(company), 社員(employee), ダメ(not good), 現場(worksite), 無い(none) | ダメ(not good), 会社(company), 現場(worksite), 責任(responsibility), カメラ(camera) | 3.83% |
| 13 | Customer Interaction | お客様(customer), 客(customer), 店(store), かける(speak), 対応(response) | お客様(customer), 汚い(dirty), かける(speak), 利用(use), 接客業(service industry) | 3.80% |
| 14 | Digital Communication | 電話(phone), 連絡(contact), 来る(come), 最悪(worst), 出勤(attendance) | 連絡(contact), 電話(phone), メール(email), 送る(send), ビデオ(video) | 3.78% |
| 15 | Role Assignment | 仕事(job), ポジション(position), 洗い場(dishwashing area), 担当(in charge), 教える(teach) | バックヤード(backyard), 軍艦(gunkan maki), 洗い場(dishwashing area), うどん(udon), ネタ(sushi toppings) | 3.70% |
| 16 | Payroll and Interview | 給料(salary), 面接(interview), 書く(write), 制服(uniform), 残業(overtime) | 制服(uniform), もらえる(can receive), 靴(shoes), 面接(interview), 定時(regular time) | 3.54% |
| 17 | Break Time Management | 食べる(eat), 休憩(break), 確認(confirm), トイレ(toilet), 忙しい(busy) | 食べる(eat), トイレ(toilet), 手洗い(hand washing), 賄い(meal), 摩可(possible) | 2.33% |
| 18 | Low Wage | 最低賃金(minimum wage), 時給(hourly wage), 安い(cheap), 労働(labor), 見合う(suitable) | 労働(labor), 賃金(wage), オーナー(owner), 最低賃金(minimum wage), 見合う(suitable) | 2.28% |
| 19 | Corporate Employee | 社員(employee), 店舗(store), 業務(task), 当たり前(usual), アルバイト(part-time job) | 発注(order), 業務(task), 社員(employee), トラブル(trouble), 頻繁(frequent) | 2.07% |
| 20 | Multinational Corporations Careers | マネージャー(manager), クルー(crew), 店舗(store), トレーナー(trainer), 仕事(job) | クルー(crew), トレーナー(trainer), マネージャー(manager), 信じる(believe), 星(star) | 2.03% |

Unique to this topic analysis were subjects reflecting the organization of chain stores (Store Manager (Topic #1), Company (Topic #12), Corporate Employee (Topic #19)), part-time jobs for students (Student Part-Time Jobs (Topic #6)), interviews (Payroll and Interview (Topic #16)), digital communication (Digital Communication (Topic #14)), and kitchen or other restaurant-related workplaces (Work Area Dynamics (Topic #10), Role Assignment (Topic #15)). Part-time jobs, having shorter employment durations than full-time positions, may more frequently involve topics like hiring interviews. Digital Communication (Topic #14) emerged as a topic of dissatis-faction, possibly due to a desire for a clearer separation between work and personal time.

Although a variety of topics related to the organization of chain stores and part-time employees were detected, the only topic related to customer interaction in this analysis was Customer Interaction (Topic #13). There were fewer instances of customer interaction topics such as complaints or gratitude than expected. It's possible that topics related to customer interactions are not impactful enough to be prominently mentioned in reviews by chain store employees.

## Topics proportion and employee satisfaction

By examining the relationship between *Employee satisfaction* scores and the topics that are more likely to emerge, it is possible to identify key factors that are important to part-time employees in chain restaurants. For this analysis, the author employed the 'estimateEffect' function from the STM package in R [11]. This function allows for estimating coefficients in the prevalence Regression Formula 1 and enables hypothesis testing on these coefficients. In this section, the result of Model 1,

$$
\begin{aligned}
prevalence_{d,k} \quad = \quad & \beta_0 + \beta_{ES} \times ES_d + \beta_{days} \times s(days_d) \\
& + \beta_{category} \times category_d + \epsilon_d,
\end{aligned}
\qquad \text{(Model 1)}
$$

is estimated and visualized. If the coefficient for a rating is significantly positive (or negative), it indicates that as the rating score increases (or decreases), the likelihood of that particular topic appearing also increases.

Based on the estimated coefficients, the author generated Fig 2, which summarizes the prevalence difference of 20 topics between reviews rated 2 (considered as relatively Bad review) and 4 (considered as relatively Good review) in each HRM factors. To generate outputs using the 'estimateEffect' function, it is necessary to select two values for comparison among the discrete values. The author chose to examine reviews with ratings of 2 and 4, as these are less likely to represent extreme opinions. Similar results were observed when the author visualized reviews rated 1 and 5. The x-axis displays the difference in prevalence scores between the two rating values. The other variables, such as s(days), and categories are held at their sample median, and then the difference in the prevalence of each topic between *Employee satisfaction* 2 and 4 is calculated [11]. The greater the distance of the points from the value of x = 0, the larger the difference in prevalence, indicating that a particular topic is more likely to occur in one of the groups. Suppose the prevalence of a topic positively correlates with the *Employee satisfaction*. In that case, it tends to appear more in the group with a 4 rating. In contrast, if it negatively correlates, it tends to appear more in the group with a 2 rating, thus intuitively showing the correlation between prevalence and rating values in the figure.

The left side of Fig 2 shows the topics frequently appearing in reviews with low *Employee satisfaction*. Newcomer Integration (Topic #7) and Work Stress (Topic #11) are strongly related to low *Employee satisfaction*. These topics seem to directly reflect low *Employee satisfaction* since they contain the word 'quit' (やめる or 辞める), which indicates turnover intention. Topics such as Digital Communication (Topic #14), Company (Topic #12), Store Manager (Topic #1), Labor Cost Reduction (Topic #8), and Payroll and Interview (Topic #16) appear next in frequency in reviews with low *Employee satisfaction*. These topics suggest possible causes of low *Employee satisfaction*. Particularly in the case of part-time workers in chain stores, the impact of digital communication outside work hours and issues related to interviews during the hiring process are unique factors not commonly found in general studies on employee reviews.

The right side of Fig 2 shows topics frequently appearing in reviews with high *Employee satisfaction*. However, given that this dataset contains a larger number of reviews with low *Employee*

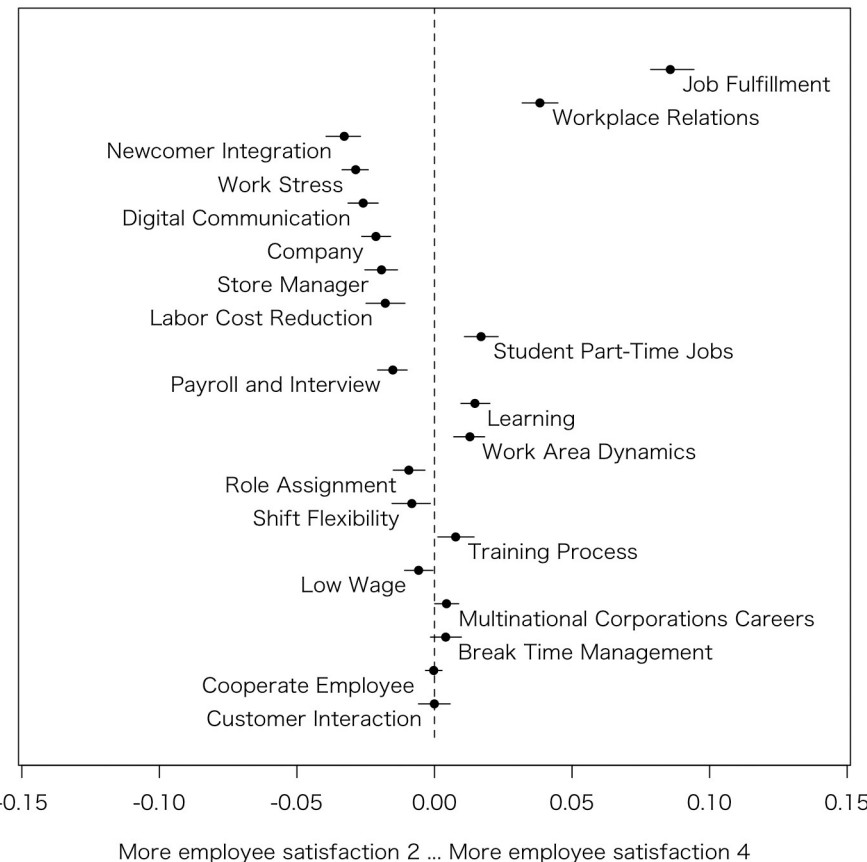

More employee satisfaction 2 … More employee satisfaction 4

**Fig 2. Topic proportion comparison (Employee satisfaction 2 vs. Employee satisfaction 4).**

*satisfaction*, it might be easier to interpret these topics as being less prevalent in reviews with low *Employee satisfaction*. Topics like Job Fulfillment (Topic #3) and Workplace Relations (Topic #2), which directly include words like 'fun' and 'good', seem to directly indicate high *Employee satisfaction*. Student Part-Time Jobs (Topic #6) may imply a demographic composition of similar members, such as students, which could contribute to higher *Employee satisfaction*. Topics like Learning (Topic #9) and Work Area Dynamics (Topic #10) do not appear in negative reviews. Analyzing the words constituting these topics, Learning (Topic #9) refers to the challenges of memorizing the menu, and Work Area Dynamics (Topic #10) refers to being busy in the kitchen. However, since these are not mentioned by employees with low satisfaction, improvements in these areas might not directly lead to enhanced *Employee satisfaction*.

## Topics proportion and four HRM ratings

Next, we examine the results of Model 2; further analysis was conducted on topics related to *Compensation satisfaction*, *Workplace environment*, *Motivation*, and *Interpersonal relationships*. The formula is as follows:

$$
\begin{aligned}
prevalence_{d,k} \quad = \quad & \beta_0 + \beta_{CS} \times CS_d + \beta_{WE} \times WE_d + \beta_M \times M_d + \beta_{IR} \times IR_d \\
& + \beta_{days} \times s(days_d) + \beta_{category} \times category_d + \epsilon_d
\end{aligned}
\quad \text{(Model 2)}
$$

In Model 2, there were no substantial differences in the topics that emerged, so the same procedure in Section 0 and 0 was followed for selecting and naming the number of topics. In this chapter, we will only discuss the results of the 'estimateEffect'.

Similarly to Model 1, based on the estimated coefficients, the author generated Figs 3–6, which summarizes prevalence difference of 20 topics between reviews rated 2 (considered as relatively Bad review) and 4 (considered as relatively Good review) in each HRM factors. The x-axis shows the variance in prevalence scores for the specific HRM factor under consideration. The other variables, such as s(days), categories, and the rest of 3 HRM factors, are held at their sample median, then calculate the difference in the prevalence of each topic between the HRM factor values 2 and 4 [11]. Therefore, by removing the influence of other HRM factors that are correlated, it becomes possible to observe topics exclusively related to that particular HRM factor. For instance, although *Workplace environment* is highly correlated with *Interpersonal relationships*, the graph specifically examines the differences in topic prevalence due to variations in *Workplace environment* when *Interpersonal relationships* is at its median value.

**Compensation satisfaction.**  In Fig 3, the author first identifies the topics discussed by those dissatisfied with their compensation. These include Low Wage (Topic #18), Labor Cost Reduction (Topic #8), Company (Topic #12), and Payroll and Interview (Topic #16). These topics are believed to relate to the amount of the salary, staffing shortages due to salary, and dissatisfaction with the companies paying these salaries.

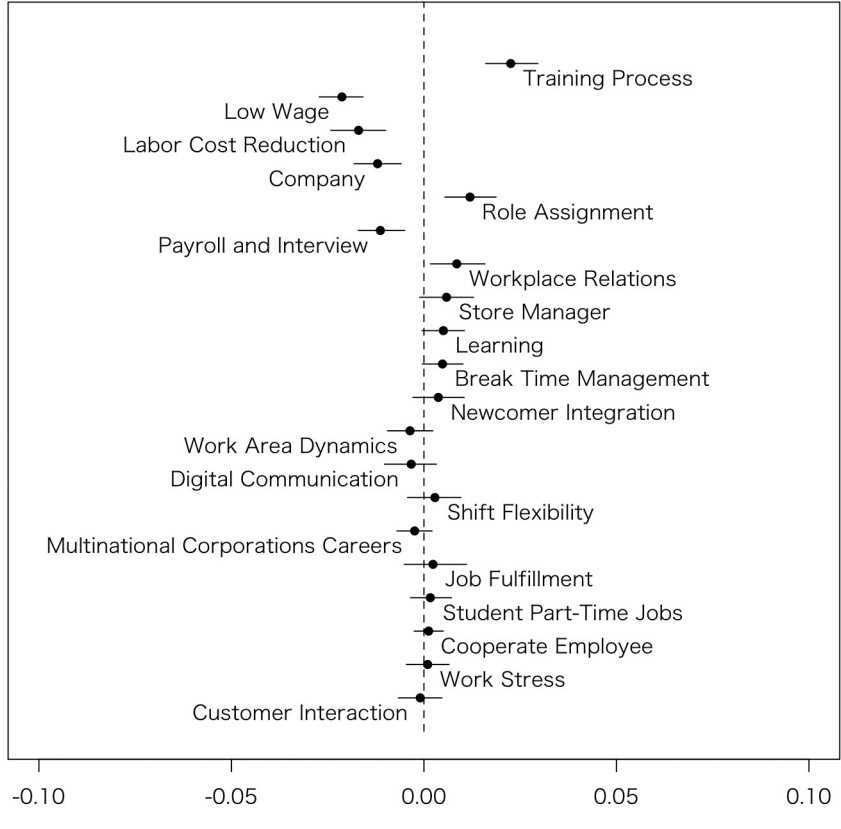

More compensation satisfaction 2 … More compensation satisfaction 4

**Fig 3. Topic proportion comparison (Compensation satisfaction 2 vs. Compensation satisfaction 4).**

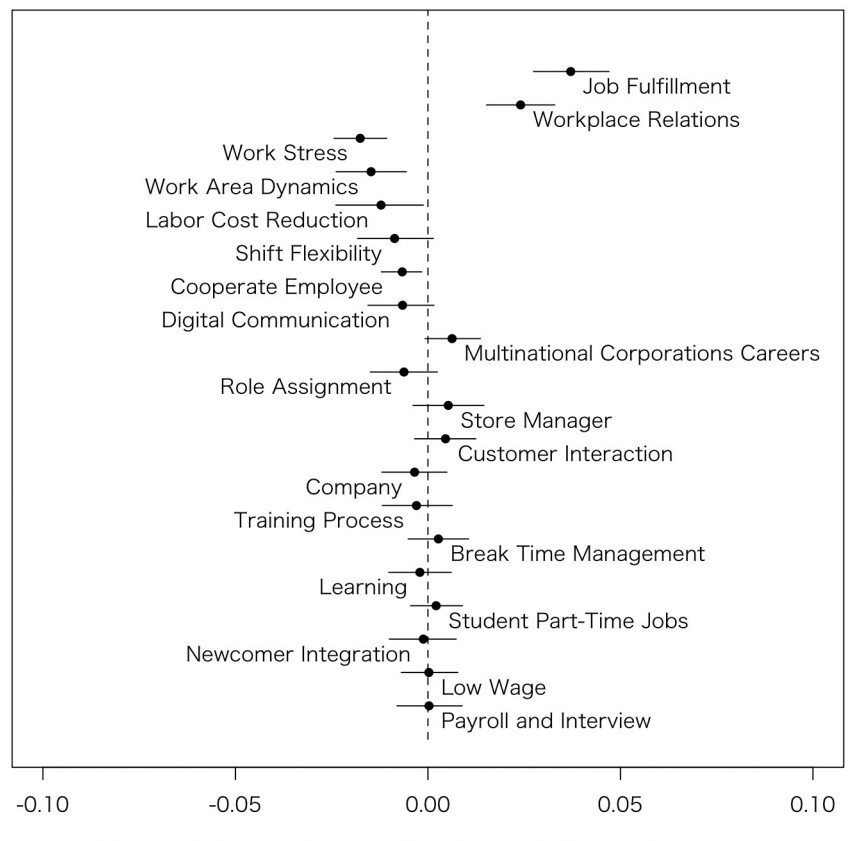

**Fig 4. Topic proportion comparison (Workplace environment 2 vs. Workplace environment 4).**

On the other hand, topics that do not indicate dissatisfaction with compensation include Training Process (Topic #5), Role Assignment (Topic #15), and Workplace Relations (Topic #2). For instance, students who join a job not for the salary but for the work experience, such as part-time work, may not express dissatisfaction with their pay and instead may discuss aspects related to the Training Process (Topic #5).

**Workplace environment.**   In Fig 4, the topics that emerge when the *Workplace environment* is rated low include, firstly, Work Stress (Topic #11), which directly represents difficult situations. Other topics like Work Area Dynamics (Topic #10) and Labor Cost Reduction (Topic #8) are also mentioned, which seem to depict the hardships of work. Work Area Dynamics (Topic #10) may refer to the challenges of working in a kitchen, while Labor Cost Reduction (Topic #8) could imply increased workloads per person due to staff shortages resulting from cost-cutting measures.

Conversely, topics not observed when scores are low include Job Fulfillment (Topic #3), which simply represents the enjoyment of work, and Workplace Relations (Topic #2), indicating positive interpersonal relationships at the workplace. In Fig 2, the topics that appear when *Employee satisfaction* is high are similar to those observed in the context of *Workplace environment*. This suggests that *Employee satisfaction* and *Workplace environment* may be closely related concepts for part-time employees.

**Motivation.**   In Fig 4, when *Motivation* is low, one of the topics that emerges is Labor Cost Reduction (Topic #8), similar to the cases of *Compensation satisfaction* and *Workplace*

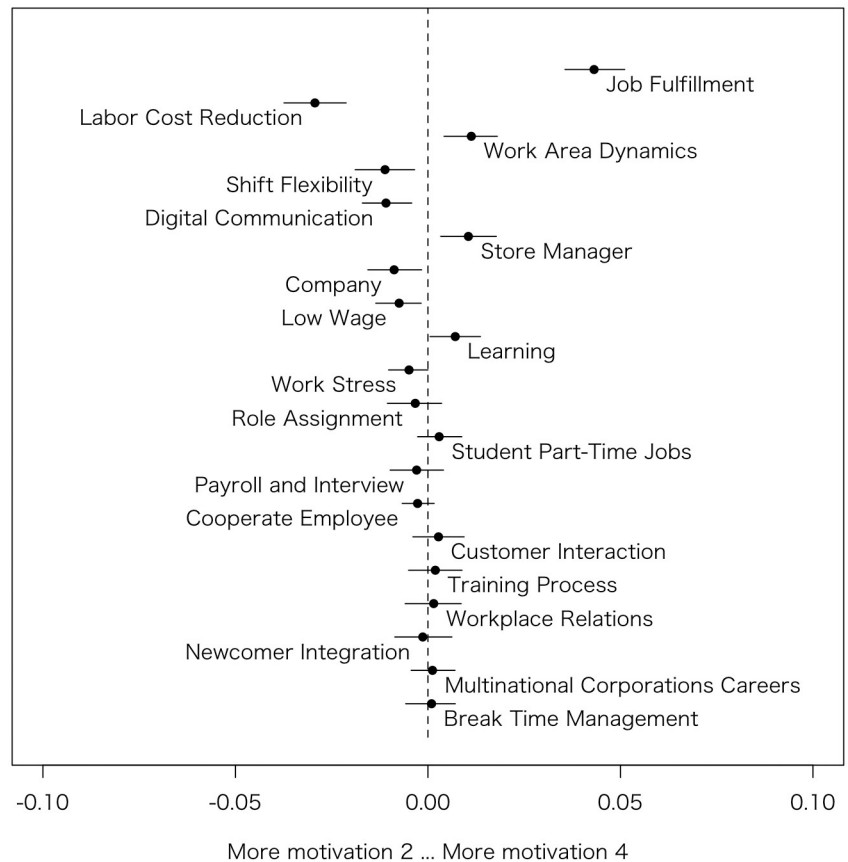

**Fig 5. Topic proportion comparison (Motivation 2 vs. Motivation 4).**

*environment.* This may include increased workload per individual due to solo operations, as well as reduced salaries resulting from restricted shifts to control labor costs. Shift Flexibility (Topic #4) could also be lowering *Motivation* for similar reasons. Digital Communication (Topic #14) might be decreasing *Motivation* due to after-hours contact, blurring the lines between work and personal time. On the other hand, those with high *Motivation* are discussing Job Fulfillment (Topic #3). Other topics, such as Work Area Dynamics (Topic #10), which were mentioned by those with a lower *Workplace environment* rating, might actually be suitably challenging for those with high *Motivation*.

**Interpersonal relationships.** In Fig 6, the score for internal relationships is associated with the emergence of various topics. Topics discussed by those with low internal relationship scores include Newcomer Integration (Topic #7) and Training Process (Topic #5), which relate to relationships with supervisors and colleagues during new employee training. In part-time jobs, since these are not long-term positions, the initial training experience is likely to be memorable. Store Manager (Topic #1), being the direct supervisor of part-time workers, is likely to be a source of various issues.

Conversely, those with low *Interpersonal relationships* scores do not discuss Labor Cost Reduction (Topic #8). This could be because cost-cutting measures lead to less overlap in shifts with other colleagues, reducing the likelihood of interpersonal issues. However, since this is also related to low *Workplace environment* and *Motivation* scores, it is not a welcome topic despite the absence of interpersonal issues. Regarding Work Area Dynamics (Topic #10),

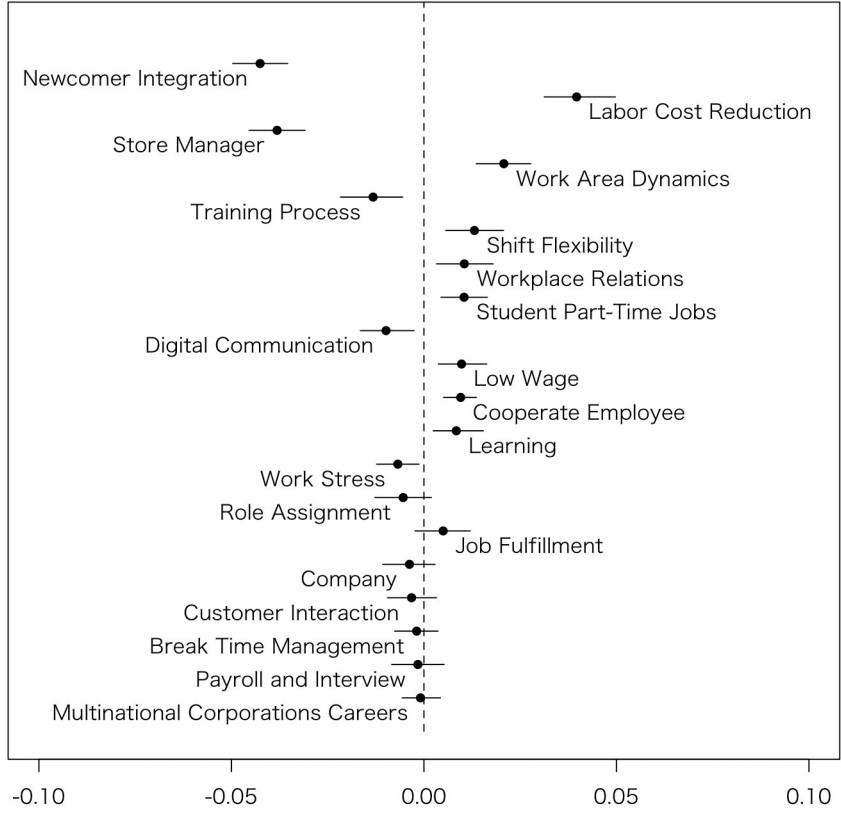

**Fig 6. Topic proportion comparison (Interpersonal relationships 2 vs. Interpersonal relationships 4).**

interpersonal issues can arise regardless of workplace arrangement, which might explain the lack of mention in low-score reviews. Shift Flexibility (Topic #4) is likely mentioned when interpersonal relations are good, as it allows for easier mutual adjustments of shifts. Additionally, Student Part-Time Jobs (Topic #6) might lead to fewer interpersonal issues as students are demographically similar to each other.

## Discussion and conclusion

This study explores the factors influencing employee satisfaction among chain restaurant part-time workers, using text data from review sites alongside employee satisfaction-related HRM factor scores (e.g., compensation satisfaction, workplace environment, motivation, interpersonal relationships, and employee satisfaction). A notable contribution of this research is the application of STM (Structured Topic Modeling) to analyze chain restaurant part-time employee reviews with HRM factor scores. This technique has revealed the types of content that part-time employees are more likely to express when dissatisfied.

The study identified 20 distinct topics from Japanese restaurant chain employee reviews. Some topics are unique to part-time employees in chain restaurants and were not observed in previous studies using general employee reviews, reflecting the organization of chain stores: topics such as store manager, company, corporate employee, part-time work, interview, digital communication, and restaurant work-related aspects. These offer a nuanced discussion in the

context of existing service research literature. Conversely, topics traditionally associated with employee satisfaction, such as benefits and work-life balance, and career advancement, were less frequently found in chain restaurant part-time employee reviews.

Furthermore, this study was able to identify which topics employees prioritize when they are dissatisfied, a novel approach in research on restaurant part-time workers. Common topics emerging when employee satisfaction was low included newcomer integration and work stress, but uniquely in this study, digital communication and payroll and interview were also significant. This suggests a strong dislike for after-hours contact via email and potential benefits in improving the payroll and interview process, reflecting previous findings on the lack of job description clarity compared to full-time positions [2].

Additionally, the study examined the relationship between the prevalence of topics and the scores of four key components constituting employee satisfaction (compensation satisfaction, workplace environment, motivation, interpersonal relationships), clarifying the content discussed by employees when scores are low in these areas. This helps identify which areas to prioritize for improvement when certain dissatisfaction is known. For example, while the training process did not show a strong correlation with employee satisfaction, it was more frequently discussed when interpersonal relationships were low and less so when compensation satisfaction was low. Employee training is important not so much for its content, but rather for how and by whom it is conducted, with attention paid to interpersonal relationships. Additionally, it was found that labor cost reduction and shift flexibility, likely emerging due to staff shortages, are particularly related to motivation. In addressing employee motivation problems, it seems more important to ensure adequate staffing levels rather than focusing on employee training. Another finding of this study is that among the topics related to customers was only one topic and its prevalence did not correlate with any of the four HRM factors and appeared uniformly across all areas. This suggests that the presence of customers is less likely to be a strong factor in causing dissatisfaction or joy among part-time employees.

This study is a pioneering application of STM in analyzing part-time employees' word-of-mouth in the Japanese service industry. It highlights the unique perspectives of part-time employees, a focus that is relatively uncommon in existing research. The study not only provides insights into employee satisfaction in Japan but also sets the stage for future cross-cultural analyses in the service sector. These findings could not have been understood through the traditional surveys conducted on part-time employees, where questions were predetermined. From a managerial standpoint, the study provides deeper insights for managers on where to improve to encourage employee satisfaction of part-time employees in chain restaurants.

However, the study has several limitations. The data used were derived from an online review platform that anyone could post. Previous research has shown that extreme opinions are more likely to be posted [28], and this study found a particular abundance of reviews focusing on dissatisfaction. While the abundance of dissatisfaction reviews aligns with the primary objective of the study, to comprehensively understand what aspects of management and job content are appreciated by part-time employees, a different methodological approach is needed. Specifically, analyzing positive stories about employee satisfaction requires using data from platforms with some incentive to post reviews. For instance, on employee review sites used for job hunting, where users must post reviews to see others' reviews due to platform rules, there would be an incentive to post reviews even if they are not extreme opinions. Focusing on enhancing positive aspects, in addition to addressing dissatisfaction, is essential for improving the retention rate of part-time employees.

Another limitation is the reliance on data from a single Japanese platform, potentially limiting its global applicability. Japanese workplace culture is often characterized by long working

hours, an emphasis on interpersonal relationships, and a high expectation of commitment to the company [37–39]. Since this study focused on part-time workers with set working hours, there was no discussion of long working hours or overtime. However, many topics related to interpersonal relationships emerged, and the connection to relationship scores might reflect the influence of Japanese workplace culture. While studies using employee reviews from other countries [9] also identified topics related to coworkers and supervisors, indicating that these are somewhat universal topics, comparative studies are needed to examine the relative frequency of these topics. Additionally, although not directly appearing in the topics, unique aspects of the Japanese service industry, such as 'Omotenashi' [40] and high service quality relative to cost [41], might also affect the generalizability of the results. Further research should expand to diverse datasets and cross-cultural comparisons, aiding both Japanese and international companies in developing culturally informed management strategies.

This study analyzed the relationship between review topics and the satisfaction of part-time employees in the restaurant industry, which faces a labor shortage, as part of a multifaceted approach to reducing turnover rates. The analysis conducted in this study is expected to serve as a starting point for further empirical research on improving turnover rates by addressing the dissatisfaction mentioned in reviews. By combining review analysis with objective indicators such as profitability, company size, and turnover rate, it is possible to identify the types of companies and management systems that are more prone to specific dissatisfaction, which could result in more effective management policies tailored to the characteristics of each company's part-time workforce. It is hoped that this study will inspire each company to address dissatisfaction from various angles and secure highly satisfied and stable part-time employees, thereby providing consistent service quality, and contributing to further improvement in the industry.

## Supporting information

**S1 Table. Topics and labels of Model 2.** The primary distinction between Models 1 and 2 lies in the proportions of topics, with no significant differences otherwise.
(PDF)

## Acknowledgments

The "Minhyo Review Dataset" was provided by meisterstudio, Inc. via IDR Dataset Service of National Institute of Informatics.

## Author Contributions

**Conceptualization:** Hiroki Takahashi.

**Data curation:** Hiroki Takahashi.

**Formal analysis:** Hiroki Takahashi.

**Funding acquisition:** Hiroki Takahashi.

**Investigation:** Hiroki Takahashi.

**Methodology:** Hiroki Takahashi.

**Project administration:** Hiroki Takahashi.

**Resources:** Hiroki Takahashi.

**Software:** Hiroki Takahashi.

**Supervision:** Hiroki Takahashi.

**Validation:** Hiroki Takahashi.

**Visualization:** Hiroki Takahashi.

**Writing – original draft:** Hiroki Takahashi.

**Writing – review & editing:** Hiroki Takahashi.

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
