## [Decision Letter · Decision Letter 0]

14 Jun 2024

PONE-D-24-15211What Do Part-Time Employees in Japanese Chain Restaurants Talk About When Dissatisfied? Applying Structural Topic Modeling to Employee ReviewsPLOS ONE

Dear Dr. Takahashi,

Thank you for submitting your manuscript to PLOS ONE. After careful consideration, we feel that it has merit but does not fully meet PLOS ONE’s publication criteria as it currently stands. Therefore, we invite you to submit a revised version of the manuscript that addresses the points raised during the review process.

We look forward to receiving your revised manuscript.

Kind regards,

Praveen Suthar, MPH

Academic Editor

PLOS ONE

“Japan Society for the Promotion of Science”

“This work was supported by JSPS KAKENHI Grant Number 23K17003. Ad[1]ditionally, the ”Minhyo Review Dataset” was provided by meisterstudio, Inc. via

IDR Dataset Service of National Institute of Informatics.”

“Japan Society for the Promotion of Science”

“I have read the journal's policy and the authors of this manuscript have the following competing interests: financial support was provided by Japan Society for the Promotion of Science, and data was provided by meisterstudio, Inc.”

4. Please note that your Data Availability Statement is currently missing the DOI/accession number of each dataset OR a direct link to access each database. If your manuscript is accepted for publication, you will be asked to provide these details on a very short timeline. We therefore suggest that you provide this information now, though we will not hold up the peer review process if you are unable.

5. Please amend either the title on the online submission form (via Edit Submission) or the title in the manuscript so that they are identical.

6. We notice that your supplementary tables are included in the manuscript file. Please remove them and upload them with the file type 'Supporting Information'. Please ensure that each Supporting Information file has a legend listed in the manuscript after the references list.

Additional Editor Comments:

Technically, the article conforms in part. I suggest adding more information to the conclusion section or having the author discuss the research's limitations. Maybe it would be better to draw attention to the cultural traits as well. Aside from these recommendations, I think the data analysis and presentation in the manuscript are reasonable, and the statistical techniques employed are sufficient.

Reviewers' comments:

Reviewer's Responses to Questions

**Comments to the Author**

1. Is the manuscript technically sound, and do the data support the conclusions?

Reviewer #1: Partly

2. Has the statistical analysis been performed appropriately and rigorously? 

Reviewer #1: Yes

3. Have the authors made all data underlying the findings in their manuscript fully available?

Reviewer #1: Yes

4. Is the manuscript presented in an intelligible fashion and written in standard English?

Reviewer #1: Yes

5. Review Comments to the Author

Reviewer #1: From a technical point of view, the article partially complies. I recommend that the conclusion section be expanded, or the author should also address the limitations of the research. Perhaps it would be better to emphasize the cultural characteristics as well. Apart from these suggestions, I consider the statistical methods used in the manuscript to be adequate, and the analysis and presentation of the data are appropriate.

6. PLOS authors have the option to publish the peer review history of their article (what does this mean?). If published, this will include your full peer review and any attached files.

Reviewer #1: No

---

## [Author Response · Author response to Decision Letter 0]

25 Aug 2024

I have addressed all the comments in the "Response to Reviewers" file.

---

## [Decision Letter · Decision Letter 1]

24 Oct 2024

What Do Part-Time Employees in Japanese Chain Restaurants Talk About When Dissatisfied? Applying Structural Topic Modeling to Employee Reviews

PONE-D-24-15211R1

Dear Dr. Takahashi,

We’re pleased to inform you that your manuscript has been judged scientifically suitable for publication and will be formally accepted for publication once it meets all outstanding technical requirements.

Kind regards,

prof. Katarzyna Piwowar-Sulej

Academic Editor

PLOS ONE

---

## [Editor Report · Acceptance letter]

22 Nov 2024

PONE-D-24-15211R1 

PLOS ONE

Dear Dr. Takahashi, 

I'm pleased to inform you that your manuscript has been deemed suitable for publication in PLOS ONE. Congratulations! Your manuscript is now being handed over to our production team.

Kind regards, 

on behalf of

Professor Katarzyna Piwowar-Sulej 

Academic Editor

PLOS ONE